# Study on Optimization Method for InSAR Baseline Considering Changes in Vegetation Coverage

**DOI:** 10.3390/s24154783

**Published:** 2024-07-23

**Authors:** Junqi Guo, Wenfei Xi, Zhiquan Yang, Guangcai Huang, Bo Xiao, Tingting Jin, Wenyu Hong, Fuyu Gui, Yijie Ma

**Affiliations:** 1Faculty of Geography, Yunnan Normal University, Kunming 650500, China; junqiguo@ynnu.edu.cn (J.G.); 240016@ynnu.edu.cn (B.X.); 2323130106@ynnu.edu.cn (T.J.); 2323130105@ynnu.edu.cn (W.H.); 2323130102@ynnu.edu.cn (F.G.); 2243206000079@ynnu.edu.cn (Y.M.); 2Yunnan Key Laboratory of Plateau Geographic Processes and Environmental Change, Kunming 650500, China; 3Key Laboratory of Early Rapid Identification, Prevention and Control of Geological Diseases in Traffic Corridor of High Intensity Earthquake Mountainous Area of Yunnan Province, Kunming 650093, China; 4Faculty of Public Safety and Emergency Management, Kunming University of Science and Technology, Kunming 650093, China; 5Key Laboratory of Geological Disaster Risk Prevention and Control and Emergency Disaster Reduction of Ministry of Emergency Management of the People’s Republic of China, Kunming University of Science and Technology, Kunming 650093, China; 6Guizhou Geological Survey Institute, Guiyang 550081, China; guangcai2020@163.com

**Keywords:** InSAR, baseline optimization, vegetation coverage, Yuanmou dry-hot valley

## Abstract

Time-series Interferometric Synthetic Aperture Radar (InSAR) technology, renowned for its high-precision, wide coverage, and all-weather capabilities, has become an essential tool for Earth observation. However, the quality of the interferometric baseline network significantly influences the monitoring accuracy of InSAR technology. Therefore, optimizing the interferometric baseline is crucial for enhancing InSAR’s monitoring accuracy. Surface vegetation changes can disrupt the coherence between SAR images, introducing incoherent noise into interferograms and reducing InSAR’s monitoring accuracy. To address this issue, we propose and validate an optimization method for the InSAR baseline that considers changes in vegetation coverage (OM-InSAR-BCCVC) in the Yuanmou dry-hot valley. Initially, based on the imaging times of SAR image pairs, we categorize all interferometric image pairs into those captured during months of high vegetation coverage and those from months of low vegetation coverage. We then remove the image pairs with coherence coefficients below the category average. Using the Small Baseline Subset InSAR (SBAS-InSAR) technique, we retrieve surface deformation information in the Yuanmou dry-hot valley. Landslide identification is subsequently verified using optical remote sensing images. The results show that significant seasonal changes in vegetation coverage in the Yuanmou dry-hot valley lead to noticeable seasonal variations in InSAR coherence, with the lowest coherence in July, August, and September, and the highest in January, February, and December. The average coherence threshold method is limited in this context, resulting in discontinuities in the interferometric baseline network. Compared with methods without baseline optimization, the interferometric map ratio improved by 17.5% overall after applying the OM-InSAR-BCCVC method, and the overall inversion error RMSE decreased by 0.5 rad. From January 2021 to May 2023, the radar line of sight (LOS) surface deformation rate in the Yuanmou dry-hot valley, obtained after atmospheric correction by GACOS, baseline optimization, and geometric distortion region masking, ranged from −73.87 mm/year to 127.35 mm/year. We identified fifteen landslides and potential landslide sites, primarily located in the northern part of the Yuanmou dry-hot valley, with maximum subsidence exceeding 100 mm at two notable points. The OM-InSAR-BCCVC method effectively reduces incoherent noise caused by vegetation coverage changes, thereby improving the monitoring accuracy of InSAR.

## 1. Introduction

InSAR technology is an active remote sensing technique that utilizes radar systems to emit microwave signals and record the returning echoes. It determines surface elevation or deformation by analyzing the phase differences between two or more radar beams collected from repeated orbits over the same area [1,2,3,4]. This capability allows InSAR to process SAR images from multiple time points, generate several interferograms, and gather time-series deformation data, thereby capturing the temporal evolution of the Earth’s surface [5,6,7]. In comparison to traditional field measurement techniques and optical remote sensing image monitoring technologies, InSAR overcomes several limitations such as hazardous working conditions and susceptibility to adverse weather conditions [8,9,10], enabling extensive and continuous monitoring over long periods. Furthermore, with the ongoing launch of more radar satellites and the increasing availability of various “cloud + edge” SAR data processing platforms, InSAR technology continually enriches its data sources and processing methods. This advancement significantly boosts its potential and unique advantages in various research fields, including landslide identification [11,12], glacier displacement monitoring [13], urban deformation monitoring [14], and more. However, performing interferometric processing on SAR images from different periods can result in a baseline network composed of interferometric pairs of varying quality, some of which may exhibit poor coherence [15,16,17]. The quality of the interferometric pairs is crucial for the high-precision inversion of surface deformation. Selecting interferometric pairs of poor quality or in limited numbers can reduce the accuracy of InSAR deformation inversion, while selecting a larger number of interferometric pairs can increase computational costs without necessarily achieving high-precision surface deformation. Therefore, optimizing the interferometric baseline network is a key area of research to enhance the accuracy of InSAR monitoring.

Currently, there are three main categories of InSAR baseline optimization methods. The first category involves manually removing interferometric pairs with poor coherence, a process guided by expert experience [18,19]. This approach, however, is time-consuming, labor-intensive, and impractical for processing large volumes of long time-series InSAR data; moreover, it introduces a level of subjectivity due to its reliance on expert knowledge. The second category involves applying threshold constraints to both temporal and spatial baselines [20,21]. The effectiveness of these methods largely depends on the settings of the temporal and spatial baseline thresholds. Setting the threshold too low can reduce the number of connected interferometric pairs, leading to a significant loss of deformation information. Conversely, setting the threshold too high can result in an increase in incoherent pairs in both time and space, thereby degrading the quality of the interferometric pairs. As a result, this approach shows considerable variability, and the determination of thresholds often heavily relies on past experiences. The third category utilizes the average coherence coefficient as a threshold to filter interferometric pairs, known as the average coherence threshold method [22]. This method can lead to discontinuities in the interferometric baseline when there is significant seasonal variation and a wide range of coherence among interferometric pairs, potentially preventing accurate inversion of surface deformation data.

While all three methods have been effective in InSAR baseline optimization research, they overlook a crucial factor that significantly influences InSAR coherence variation: vegetation coverage. Vegetation—including plant communities such as forests, shrubs, and grasslands—is a critical component of the geographical environment, playing a vital role in surface energy exchange and global water cycles [23]. Dense vegetation can cause radar signals to undergo multiple reflections or scattering between leaves and branches, which complicates the signals’ ability to reach the ground. Moreover, the moisture and other components within vegetation can absorb radar signals, leading to signal attenuation. Variations in the growth and density of vegetation also weaken radar signals [24], and different levels of vegetation coverage impact radar signals to varying degrees [25,26]. Consequently, accounting for changes in vegetation coverage is a key issue in optimizing the InSAR baseline.

Addressing the issues previously mentioned, this study introduces an optimization method for the InSAR baseline considering changes in vegetation coverage (OM-InSAR-BCCVC). This method has been validated in the Yuanmou dry-hot valley through rigorous experimental verification. Initially, leveraging the Google Earth Engine (GEE) platform and Sentinel-2 data, this study computes a monthly time series of Fraction Vegetation Coverage (FVC) from January 2021 to May 2023. The overall average FVC serves as a threshold to categorize the months into periods of high- and low-vegetation coverage. Following this, this study employs the Global Atmospheric Correction Online Service (GACOS) to perform atmospheric corrections on all generated interferometric pairs. These pairs are then further subdivided based on the month of acquisition into those associated with high vegetation coverage and those with low. Using the average coherence value as a cutoff, pairs falling below this threshold in both categories are excluded. The Small Baseline Subset InSAR (SBAS-InSAR) inversion process follows, tailored specifically to each vegetation category. Finally, after applying geometric distortion region masking, detailed radar line of sight (LOS) surface deformation data for the Yuanmou dry-hot valley from January 2021 to May 2023 are obtained. Validation of the results is conducted through landslide identification using optical images from Google. The OM-InSAR-BCCVC method proposed in this study effectively reduces InSAR incoherent noise triggered by changes in vegetation coverage, thereby enhancing the accuracy of InSAR monitoring.

## 2. Study Area

### 2.1. Geographical Location and Topographical Features

This study is conducted in the Yuanmou dry-hot valley, located in the Chuxiong Yi Autonomous Prefecture, Yunnan Province, China (25°21′ to 26°6′ N, 101°35′ to 102°6′ E), as shown in Figure 1. Positioned in the northern part of the Yunnan Plateau and the eastern section of the Hengduan Mountains, the Yuanmou dry-hot valley lies downstream of the Longchuan River, a major tributary of the Jinsha River [27,28]. The terrain is a typical high-mountain canyon, surrounded by high elevations with a central valley that dips to a lower altitude. Elevations range from 868 m to 2838 m, with a relative height difference of 1970 m. The region is crossed by the Yuanmou fault zone, which runs north to south, characterized by numerous fault planes and fractured zones [29], creating diverse landforms such as basins, hills, and plateaus [30].

### 2.2. Climatic Characteristics

The Yuanmou dry-hot valley’s climate is heavily influenced by its enclosed basin topography and significant relative height differences, which restrict sources of water and vapor. The climate is shaped by the southwest monsoon and continental air currents, leading to distinct dry and wet seasons. Annually, the valley receives about 624 mm of precipitation, with roughly 90% of it falling between June and October. The dry season lasts from November to May, during which only about 10% of the annual precipitation occurs. The annual evaporation rate is approximately 3848 mm, six times that of the precipitation, resulting in a severe water–heat imbalance and a characteristic dry-hot climate [31,32].

### 2.3. Vegetation Features

Adapted to the harsh dry-hot climate, the vegetation in the Yuanmou dry-hot valley presents a unique community appearance and composition. There are significant seasonal changes: vegetation flourishes during the wet season and becomes sparse in the dry season. The dominant tree species are Pinus yunnanensis, Phyllanthus emblica, and Ziziphus mauritiana. Common shrubs include Acacia farnesiana, Vitex negundo, and Osteomeles anthyllidifolia, while the primary herbaceous species is Heteropogon contortus [33,34]. This plant community is well-adapted to the long-term arid conditions of the valley.

## 3. Data Description

The data used in this study include the following: ① C-band Sentinel-1A ascending orbit data from the European Space Agency (ESA) Copernicus program, consisting of 74 scenes in Interferometric Wide (IW) mode, with Single Look Complex (SLC) images in VV polarization. The spatial resolution is 5 m × 20 m, covering the time span from January 2021 to May 2023. These data were utilized for interferometric stacking to obtain surface deformation information in the Yuanmou dry-hot valley. They can be freely downloaded from https://search.asf.alaska.edu/#/, accessed on 9 January 2024. ② Optical remote sensing images from the European Space Agency (ESA) Copernicus Sentinel-2 program, with a spatial resolution of 10 m. These images were used to calculate the Fraction Vegetation Coverage (FVC) in the study area and can be freely downloaded from https://scihub.copernicus.eu, accessed on 9 January 2024. ③ Copernicus Sentinel POD Precision Orbit Ephemeris data, obtained from the European Space Agency (ESA) were used to enhance satellite orbit accuracy. These data can be accessed online at https://dataspace.copernicus.eu/, accessed on 12 January 2024. ④ A 30-meter resolution Digital Elevation Model (DEM) provided by the Japan Aerospace Exploration Agency (JAXA) ALOS WORLD 3D. This DEM was used to correct terrain phase effects in Sentinel-1A data processing and can be obtained online at https://www.eorc.jaxa.jp/ALOS/en/aw3d30/data/index.htm, accessed on 20 January 2024. ⑤ Global Atmospheric Correction Online Service (GACOS) data, used to correct atmospheric delay errors in interferometric pairs generated from Sentinel-1A data stacking. These data enhance InSAR accuracy and are available for free at http://www.gacos.net/, accessed on 10 January 2024. ⑥ Google satellite imagery, utilized to assist in landslide identification based on InSAR results, can be accessed at http://www.google.cn/intl/zh-CN/earth/, accessed on 20 February 2024. For more detailed information about the data, refer to Table 1.

## 4. Research Methods and Data Processing

The overall technical route of this study is shown in Figure 2, and the main technical processes include the following: (1) interference superposition and GACOS atmospheric correction based on Sentinel-1A data, (2) pixel bipartite FVC calculation based on Sentinel-2 data, and (3) baseline optimization and SBAS-InSAR surface deformation information retrieval.

### 4.1. Interferometric Superposition and GACOS Atmospheric Correction Based on Sentinel-1A Data

In theory, connecting all SAR images for interferometric stacking to generate as many interferometric pairs as possible would offer more options for baseline optimization. However, as the time interval between two SAR images increases, the temporal and spatial incoherence of the resulting interferometric pairs also increase, resulting in many unusable low-coherence pairs [35,36]. To manage data redundancy, we set a spatial baseline limit of 50% and a temporal baseline of 108 days for the 74 ascending orbit Sentinel-1A scenes from January 2021 to May 2023. This configuration yielded a total of 423 interferometric pairs, which served as the basis for InSAR baseline optimization. The baseline connectivity is illustrated in Figure 3.

Atmospheric influences on electromagnetic wave signals can cause refraction, altering the signal’s propagation path and direction during transmission [37]. In single-orbit InSAR modes, atmospheric effects on two images are very similar and can largely be canceled out through interferometric processing; however, in repeat-orbit modes, variations in atmospheric conditions during the image acquisition interval can result in significant atmospheric delay phase effects on the interferometric phase [38,39]. Thus, during interferometric stacking, we use the GACOS atmospheric correction model, developed by Professor Zhenhong Li’s team at Newcastle University [40]. This system utilizes high-resolution atmospheric numerical products from the European Centre for Medium-Range Weather Forecasts (ECMWF) with a grid resolution of 0.1° × 0.1°. It employs the Iterative Tropospheric Decomposition (ITD) model to separate the layered and turbulent components from tropospheric delay, delivering corrected images with a resolution of 90 m for Zenith Tropospheric Delay (ZTD). The ZTD formula is defined as follows [41,42]:(1)$$ZTD_{k}=T_{\left(x_{k}\right)}+L_{0}e^{-\beta {\bar{h}}_{k}}+\varepsilon_{k}$$
where *k* represents the coordinate position; *ZTDk* is the total zenith delay that combines the vertical layered component and the horizontal turbulent component; *T* denotes the turbulent signal; *x_k_* is the site coordinate vector in the local geocentric coordinate system; *L*_0_ represents the layered component delay at sea level; $\varepsilon_{k}$ denotes the remaining unmodeled residuals, including unmodeled layered and turbulent signals; and ${\bar{h}}_{k}$ is the height scale, calculated by the following formula.
(2)$${\bar{h}}_{k}=\frac{h_{k}-h_{\min}}{h_{\max}-h_{\min}}$$

We download GACOS products corresponding to each SAR image acquisition to apply atmospheric delay phase correction to the generated interferometric pairs. Notably, not all GACOS products can effectively correct every interferometric pair; in some cases, they may even increase phase delay errors [43]. The phase standard deviation (STD) is used as an indicator to assess the effectiveness of atmospheric correction. As shown in Figure 4, after GACOS correction, out of 423 interferometric pairs, the STD decreased in 362 pairs (85.6%), while the STD increased in 61 pairs (14.4%), with the maximum increment being only 6.8%. Overall, in the Yuanmou dry-hot valley area, GACOS data effectively improve the phase errors induced by atmospheric delays.

### 4.2. Pixel Dichotomy FVC Calculation Based on Sentinel-2 Data

Vegetation, a crucial component of terrestrial ecosystems, acts as an indicator of ecosystem changes. Vegetation coverage, which quantitatively represents vegetation growth, plays an increasingly significant role in regional and global studies related to hydrology, meteorology, ecology, and other fields. Generally, vegetation coverage is defined as the percentage of the ground area covered by vegetation—this includes leaves, stems, and branches—relative to the total study area and is indicative of vegetation growth trends. A commonly used method to calculate Fractional Vegetation Coverage (FVC) is the pixel-wise binary model [44].

The pixel-wise binary model is a straightforward and practical remote sensing estimation model. It assumes that the Earth’s surface consists of pixels that contain both vegetation-covered and non-vegetation-covered portions. The calculation for the proportion of vegetation-covered area within a pixel can be represented by the following formula [45,46]:(3)$$FVC=\frac{\left(S-S_{soil}\right)}{\left(S_{veg}-S_{soil}\right)}$$
where *S* represents a single pixel information; *S_soil_* denotes soil information, that is, remote sensing pixel information without vegetation coverage; and *S_veg_* represents remote sensing pixel information with vegetation coverage.

The Normalized Difference Vegetation Index (NDVI) is also a remote sensing index derived from the spectral information received by remote sensing sensors, reflecting the vegetation status of the Earth’s surface. The NDVI value of a pixel can be expressed as information containing green vegetation cover (*NDVI_veg_*) and information without vegetation cover (*NDVI_soil_*). Following the improvement of the parameters *S_soil_* and *S_veg_* in the pixel-wise binary model using the NDVI index by Li Miaomiao [47], Equation (3) can be rewritten as follows:(4)$$FVC=\left\{\begin{array}{cc} 0 & ,NDVI\leq NDVI_{soil}\\ \frac{NDVI-NDVI_{soil}}{NDVI_{veg}-NDVI_{soil}} & ,NDVI_{soil}\leq NDVI\leq NDVI_{veg}\\ 1 & ,NDVI\geq NDVI_{veg} \end{array}\right.$$

Based on the methods mentioned above, utilizing *NDVI* as individual pixel information, and using the Google Earth Engine platform, monthly *NDVI* values for the study area were calculated from Sentinel-2 optical remote sensing images from January 2021 to May 2023. Subsequently, based on the principles of the pixel-wise binary model, the monthly vegetation coverage for the study area was further calculated, and a portion of the FVC is shown in Figure 5. The study area’s vegetation coverage for each month was classified into months with high vegetation coverage and months with low vegetation coverage using the average FVC value as the threshold, with the calculation formula as follows:(5)$$FVC_{avg}=\frac{\sum_{i=1}^{n}FVC_{i}}{n}$$
(6)$$\left\{\begin{array}{l} FVC_{i}>FVC_{avg},\mathrm{High}\;\mathrm{vegetation}\;\mathrm{cover}\;\mathrm{months}\\ FVC_{i}<FVC_{avg},\mathrm{Low}\;\mathrm{vegetation}\;\mathrm{cover}\;\mathrm{months} \end{array}\right.$$

High vegetation coverage months were identified as July, August, September, October, November, and December in 2021, along with July, August, October, and November in 2022. The remaining months were classified as periods of low vegetation coverage.

### 4.3. Baseline Optimization and Inversion of SBAS-InSAR Surface Deformation Information

Based on the 423 interferometric pairs obtained after interferometric stacking and atmospheric correction in Section 4.1, the pairs were further categorized into interferometric pairs from months with high vegetation coverage and those from months with low vegetation coverage. When there were pairs spanning multiple months (e.g., SAR images from February and April combined into one interferometric pair), the categorization was based on the comparison between the average vegetation coverage *FVC*_2,4_ of February and April and the overall average *FVC_avg_*. Among these, 192 pairs were categorized as interferometric pairs from months with high vegetation coverage, while 231 pairs were categorized as interferometric pairs from months with low vegetation coverage. 

Calculating the pixel coherence and average coherence for each interferometric pair is essential, as coherence is a crucial metric for assessing the quality of interferometric pairs. Coherence is defined by the cross-correlation function during the co-registration of two SAR images. The coherence of a pixel *P* at (*x*, *y*) can be expressed as the following:(7)$$P_{\left(x,y\right)}=\frac{\left|\sum_{x=1}^{m}\sum_{y=1}^{n}M(x,y)S^{*}(x,y)\right|}{\sqrt{\sum_{x=1}^{m}\sum_{y=1}^{n}\left|M{\left(x,y\right)}^{2}\right|\sum_{x=1}^{m}\sum_{y=1}^{n}\left|S{\left(x,y\right)}^{2}\right|}}$$
where (*x*, *y*) represents the coordinates of a pixel in the radar slant-range coordinate system; parameters *m* and *n* denote the window size for the coherence calculation; *M* and *S* represent the two SAR data acquisitions; and * denotes the complex conjugate of a given complex number. After calculating the coherence for each pixel, the average coherence of the entire interferometric pair can be computed using Equation (8):(8)$$C_{avg}=\frac{\sum_{x=1}^{l}\sum_{y=1}^{w}C_{\left(x,y\right)}}{l\times w}$$
where (*x*, *y*) represents the coordinates of a pixel in the radar slant-range coordinate system; *l* and *w* denote the length and width of the interferometric pair image, respectively; *C* represents the coherence coefficient of interferometric pairs, with a range of [0, 1]. A value of 0 indicates complete incoherence, while 1 indicates perfect coherence. A higher coherence coefficient implies greater coherence in interferometric pairs, indicating a higher similarity between the two backscattered signals. This high coherence allows for accurate reflection of the distance difference between the two echoes, leading to more precise information on ground deformation [48].

The average coherence coefficient threshold method directly uses the arithmetic mean of the coherence coefficients of all interferometric pairs as the threshold, discarding those interferometric pairs with coherence below the threshold. The discrimination formula is as follows:(9)$$\left\{\begin{array}{l} C_{i}>C_{avg},\mathrm{Preserve}\;\mathrm{image}\;\mathrm{pairs}\\ C_{i}<C_{avg},\mathrm{Delete}\;\mathrm{image}\;\mathrm{pair} \end{array}\right.$$

In this study, we considered changes in surface vegetation coverage and categorized interferometric pairs into those formed during months of high vegetation coverage and those formed during months of low vegetation coverage. We then applied Equation (9) to exclude pairs from each category accordingly. Among them, 64 pairs formed during high vegetation coverage months were excluded, and 74 pairs formed during low vegetation coverage months were excluded. The optimized baseline network is shown in Figure 6.

Subsequent SBAS-InSAR processing on the optimized interferometric pairs was performed using Sarscape5.6 software. The SBAS-InSAR technique, developed by Berardino and Lanari based on differential InSAR, is a time-series InSAR method capable of obtaining surface deformation information at the centimeter or even millimeter level [49]. Assuming the acquisition of *N* + 1 SAR images at different times for the same region and the same orbit, for the *i* (*i* = 1, 2…, *N* + 1) interferogram generated by interferometric stacking of SAR images at times TA and TB, the interferometric phase for pixels in azimuth coordinate *x* and range coordinate y can be expressed as follows:(10)$$\Delta \varphi_{i}(x,y)=\Delta \varphi_{T_{B}}(x,y)-\Delta \varphi_{T_{A}}(x,y)\approx \Delta \varphi_{i}^{def}+\Delta \varphi_{i}^{topo}+\Delta \varphi_{i}^{atm}+\Delta \varphi_{i}^{noise}$$
where $\Delta \varphi_{i}^{def}$ represents the accumulated deformation in the radar line-of-sight direction; $\Delta \varphi_{i}^{topo}$ represents the residual topographic phase in the differential interferogram; $\Delta \varphi_{i}^{atm}$ represents the atmospheric delay phase; and $\Delta \varphi_{i}^{noise}$ represents the decorrelated noise.

To obtain physically meaningful time-series deformation information, the phase in Equation (9) is expressed as the product of the average phase velocity between two acquisition times and time:(11)$$v_{i}=\frac{\varphi_{i}-\varphi_{i-1}}{T_{i}-T_{i-1}}$$

The phase value of the *i* interferogram can be expressed as follows:(12)$$\sum_{p=T_{A,i+1}}^{T_{B,i}}\left(T_{p}-T_{p-1}\right)v_{p}=\Delta \varphi_{i}$$

The integration of each time period on the time interval between the master and slave images is represented by an *M* × *N* matrix:(13)Bv=Δφ

Due to the use of a multi-master image strategy in the SBAS-InSAR technology, the matrix B becomes rank-deficient. Therefore, the Singular Value Decomposition (SVD) method is employed to calculate the generalized inverse matrix of matrix B. Subsequently, the minimum-norm solution for the velocity vector is computed. Finally, by integrating the velocities between different time periods, the deformation values for each time period can be obtained [50,51].

SAR is a side-looking imaging sensor, and the radar beam is directed toward the target with a certain angle of incidence. In the range direction, the imaging is recorded based on the sequence of reflected information from the target. Even subtle changes in elevation can result in significant distortions in the image. The local incidence angle θ and the radar line-of-sight incidence angle α play a crucial role in influencing the characteristics of the image, introducing inherent geometric distortions such as shadows and layover during the imaging process [52]. The relationship between θ and α can be expressed as follows:(14)$$\left\{\begin{array}{cc} \theta <0{}^{\circ}\; & ,Layover\\ \theta >90{}^{\circ}\; & ,Shadow\\ 0{}^{\circ}\leq \theta \leq 90{}^{\circ}\;and\;\theta -\alpha >0{}^{\circ}\; & ,Good\;visibility \end{array}\right.$$

In order to ensure the accuracy of InSAR monitoring, according to the radar satellite parameters, the radar visibility analysis of the study area was carried out based on the LSM (Layover and Shadow Map) algorithm to identify the geometric distortion area, and the results are shown in Figure 7.

From Figure 7, it can be observed that the geometric distortion regions are mainly concentrated in the northeastern part of the Yuanmou dry-hot valley. Among them, the layover area accounts for 4.4% of the entire study area, and the shadow area represents 0.2% of the total area. During the InSAR data processing, masking should be applied to the identified geometric distortion regions to ensure the accuracy of the InSAR results.

## 5. Results and Analysis

After applying GACOS atmospheric correction, OM-InSAR-BCCVC baseline optimization, and geometric distortion region masking, the radar line of sight (LOS) deformation rates for the Yuanmou dry-hot valley from January 2021 to May 2023 were obtained, as illustrated in Figure 8. The overall deformation rate ranged from −73.87 mm/yr to 127.35 mm/yr, where a positive rate indicates uplift toward the satellite and a negative rate indicates subsidence away from the satellite. Fifteen landslides and potential landslide areas were identified and numbered from 1 to 15, with detailed deformation information depicted in Figure 9.

From Figure 8, it can be observed that landslides in the Yuanmou dry-hot valley are predominantly concentrated in the northern region. This phenomenon may be attributed to the proximity of this area to the Yuanmou fault zone. The fault zone represents a location where relative movements occur between crustal plates, inducing deformation or torsion in the crust. This deformation leads to the formation of cracks and stress in rock layers, increasing the likelihood of landslide occurrences. Additionally, the convergence of the Jinsha River and the Longchuan River near this region began reservoir filling in May 2022. The rise in water levels, coupled with increased pore water pressure and seepage, reduces slope stability. The progressive failure mechanism, coupled with factors such as rainfall infiltration, contributes to the deformation of the mountainside, eventually leading to landslide development. Figure 10 shows Google Earth satellite images of the confluence of the Jinsha River and the Longchuan River in Yuanmou County in 2020 and 2022, highlighting the noticeable change in water levels. The central and southern regions, where the surface uplift areas align with the orientation of water systems, may be influenced by the composition of the soil. The Yuanmou dry-hot valley consists mostly of arid red soil with high clay content. The larger particles and loose structure of the arid red soil allow it to quickly absorb and store significant amounts of water. This soil’s capacity for water absorption and expansion leads to surface uplift; moreover, the water systems in the Yuanmou dry-hot valley are often located in low-altitude areas with relatively flat terrain. The slow flow of rivers in these areas allows sediment, such as mud and gravel, to accumulate on the riverbed and banks, contributing to surface uplift.

To further analyze the deformation characteristics of landslides in the Yuanmou dry-hot valley, taking landslides 12 and 15 as examples, the obtained deformation rate points are overlaid on Google satellite images. Selecting characteristic points A and B, the time-series deformation curves are plotted, as shown in Figure 11 and Figure 12.

From Figure 11, it can be observed that the central part of landslide creep body 12 mainly experiences subsidence deformation, with a maximum deformation rate of −40.35 mm/yr. In the valley positions on both sides of the slope, soil accumulation has occurred. Examining the time-series deformation curve for characteristic point A shows continuous subsidence from January 2021 to May 2023, with a relatively small oscillation amplitude, reaching a maximum subsidence of around 80 mm. The reason may be that the landslide creep body has a steep slope and is located close to the reservoir area and fault zone. Under the influence of water impoundment pressure and geological activity, this has resulted in the continuous slow subsidence of the landslide creep body.

The middle and lower sections of landslide creep body 15 primarily exhibit subsidence deformation, while the valley bottom area in the lower right corner shows uplift deformation, with a maximum deformation rate of −54.47 mm/year and a maximum subsidence exceeding 100 mm. Characteristic point B experiences subsidence deformation during months of low vegetation coverage (January–June 2021, January–June, September, and December 2022, and January–May 2023), and deformation remains relatively stable during months of high vegetation coverage (July–December 2021, July, August, October, and November 2022). When vegetation coverage is high, the development of vegetation and the interception of rainfall by leaves help to reduce surface erosion, resulting in a more stable deformation curve. In contrast, when vegetation coverage is low, the soil is directly exposed to rainfall, failing to effectively absorb and retain water. This is especially problematic in sloped areas where increased soil erosion leads to significant surface subsidence, causing abrupt changes in the deformation curve.

## 6. Discussion

This study introduces an InSAR baseline optimization method that accounts for changes in vegetation coverage. To evaluate the accuracy and effectiveness of the proposed method, we compared it with the average coherence threshold method and a method without baseline optimization. Figure 13 illustrates the baseline connections for the three methods. From the figure, it is evident that while the average coherence threshold method generally enhances the coherence of interferometric pairs, it also leads to baseline discontinuities that hinder the subsequent inversion of surface deformation information. This issue stems from the significant seasonal variation in vegetation coverage in the Yuanmou dry-hot valley, which causes notable fluctuations in interferometric pair coherence. Coherence is lowest in July, August, and September, and highest in January, February, and December. Consequently, the overall average coherence threshold, being much higher than that of the low-coherence months, proves insufficient for the unique conditions of the Yuanmou dry-hot valley. In contrast, the proposed OM-InSAR-BCCVC method not only improves the overall coherence of interferometric pairs but also maintains the connectivity of the interferometric baseline network.

Continuing with our comparative analysis, we further assessed the proposed method against the average coherence threshold method and the method without baseline optimization by comparing the effective interferogram ratio and root mean square error (RMSE) of surface deformation inversion using the SBAS-InSAR technique. Figure 14 illustrates these comparisons. The effective interferogram ratio refers to the proportion of interferograms that contribute to the inversion calculation out of all the interferograms during the inversion of surface deformation information using the SBAS-InSAR method. A higher effective interferogram ratio indicates that more interferograms are involved in the inversion calculation, which improves the reliability and accuracy of the monitoring results. The RMSE measures the error in surface deformation inversion by the SBAS-InSAR method, with lower values indicating better model-fitting and more accurate inversion. From Figure 14, it is clear that the use of the proposed baseline optimization method results in improvements in both the effective interferogram ratio and RMSE. To further quantify these enhancements, we calculated the overall average values for the effective interferogram ratio and RMSE, as presented in Table 2. After implementing the proposed method, the overall effective interferogram ratio increased by 17.5%, and the overall RMSE decreased by 0.5 rad. These results underscore the effectiveness of the OM-InSAR-BCCVC method in adapting to the dynamic vegetation conditions of the Yuanmou dry-hot valley; moreover, by providing a more reliable and accurate approach for InSAR monitoring in similar challenging environments, the method proposed in this study can effectively improve the accuracy of surface deformation inversion using SBAS-InSAR.

Compared to the overall average coherence in the average coherence threshold method, this study takes into account the variability in surface vegetation cover by dividing interferograms into categories based on vegetation coverage and the timing of interferogram pair combinations. The average coherence of each category is then used as a threshold to selectively filter interferogram pairs within each category. For instance, in this study, the average coherence threshold method yields an average coherence of 0.41 for all interferogram pairs. However, for interferogram pairs acquired during months with low vegetation cover, the average coherence is 0.43, while for those during months with high vegetation cover, it drops to only 0.35. This variance suggests that employing the average coherence threshold method for baseline optimization would lead to the exclusion of many high-vegetation-cover interferogram pairs, resulting in discontinuities in the interferometric baseline network. In contrast, the method without baseline optimization retains more interferogram pairs but includes a significant number of low-coherence pairs, ultimately reducing the accuracy of surface deformation inversion using SBAS-InSAR. The OM-InSAR-BCCVC method proposed in this study effectively mitigates the incoherent noise caused by changes in surface vegetation cover, thereby enhancing the precision of InSAR monitoring. This approach represents a significant refinement in InSAR technology application, focusing on adapting to dynamic environmental conditions to maintain high data integrity.

In 1989, Gabriel first introduced the concept of Differential Interferometric Synthetic Aperture Radar (D-InSAR), achieving the separation of surface deformation and topography. This innovation marked the beginning of a new era in monitoring surface deformations using InSAR technology [53]. Over the subsequent decades, rapid development in data sources and processing techniques has led to the widespread application of InSAR technology across various fields. However, challenges such as geometric distortions caused by terrain undulations, incoherent noise due to changes in vegetation cover, and atmospheric delay errors induced by variations in water vapor continue to limit the full potential of InSAR technology in many domains. In this study, we utilized the GACOS product for atmospheric delay correction of interferograms, proposed an innovative InSAR baseline optimization method considering changes in vegetation cover (OM-InSAR-BCCVC), and conducted mask processing on areas with geometric distortions to obtain accurate surface deformation information in the Yuanmou dry-hot valley. We have demonstrated the significant applicability of the OM-InSAR-BCCVC method in complex mountainous areas with substantial changes in surface vegetation coverage. However, its use may be limited in some low-latitude tropical regions and high-latitude grasslands and glacier areas. This limitation arises due to the minimal seasonal vegetation changes in tropical regions and the low or absent vegetation coverage in high-latitude grasslands and glaciers throughout the year. Nonetheless, the concept of interferometric baseline optimization proposed in this study may still hold some relevance in these regions. The ultimate goal of InSAR baseline optimization is to exclude low-coherence redundant pairs from a large number of interferometric pairs, thereby improving the accuracy of surface deformation inversion. In this study, we classified interferometric pairs based on surface vegetation coverage into two categories to achieve interferometric baseline optimization. The purpose of this classification is to reduce coherence fluctuations caused by seasonal variation factors, thereby lowering coherence dispersion and ensuring the integrity of the interferometric baseline network. In regions such as rainforests, grasslands, and glaciers, other seasonal variation factors should be considered for interferometric baseline optimization. For instance, in tropical regions, vegetation coverage can be more finely divided into high, medium, low, or more levels, and weighted averaging methods can be used for baseline optimization. Additionally, interferometric pairs can be classified and optimized based on seasonal variation factors affecting InSAR coherence, such as precipitation and climate. In high-latitude grasslands and glaciers, vegetation-induced decorrelation noise can be ignored, and factors such as seasonal weathering and erosion in grasslands and seasonal melting in glaciers should be considered for classifying interferometric pairs to achieve baseline optimization. Such an approach not only enhances the method’s applicability but also provides new insights for applying InSAR technology under different geographical and environmental conditions.

## 7. Conclusions

This study explores a new direction for InSAR baseline optimization research by considering the impact of surface vegetation coverage changes on InSAR coherence. We designed and proposed an InSAR baseline optimization method that accounts for vegetation coverage changes, termed OM-InSAR-BCCVC. First, we calculated the monthly time series of FVC (Fractional Vegetation Cover) from January 2021 to May 2023. Using the average FVC as the threshold, we categorized the months into high- and low-vegetation coverage periods. Then, we introduced GACOS products to perform atmospheric correction on all generated interferometric pairs. Based on the imaging months of the interferometric pairs, we classified them into high- and low-vegetation coverage groups. We used the average coherence coefficient as the threshold to exclude pairs below the threshold in both categories and conducted subsequent SBAS-InSAR inversion. Finally, after masking geometric distortion regions, we obtained the radar line-of-sight (LOS) surface deformation information for the Yuanmou dry-hot valley from January 2021 to May 2023 and verified landslide identification using Google optical images. The experimental results indicate the following: (1) Due to the significant seasonal variation in vegetation coverage in the Yuanmou dry-hot valley, InSAR coherence also exhibits noticeable seasonal changes. Coherence is lowest in July, August, and September and highest in January, February, and December. The average coherence coefficient threshold method is limited in the Yuanmou dry-hot valley, causing the interferometric baseline network to break. Compared with the method without baseline optimization, the OM-InSAR-BCCVC method improved the effective interferogram ratio by 17.5% and reduced the inversion error RMSE by 0.5 rad. (2) After GACOS atmospheric correction, baseline optimization, and geometric distortion region masking, the LOS surface deformation rate in the Yuanmou dry-hot valley ranged from −73.87 mm/year to 127.35 mm/year. We identified 15 landslides and potential landslides, mainly distributed in the northern region, with two characteristic points showing maximum subsidence exceeding 100 mm. The proposed method effectively mitigates the impact of seasonal vegetation coverage changes on InSAR coherence, providing new insights for applying InSAR technology in complex mountainous areas with significant surface vegetation coverage changes.

## Figures and Tables

**Figure 1 sensors-24-04783-f001:**
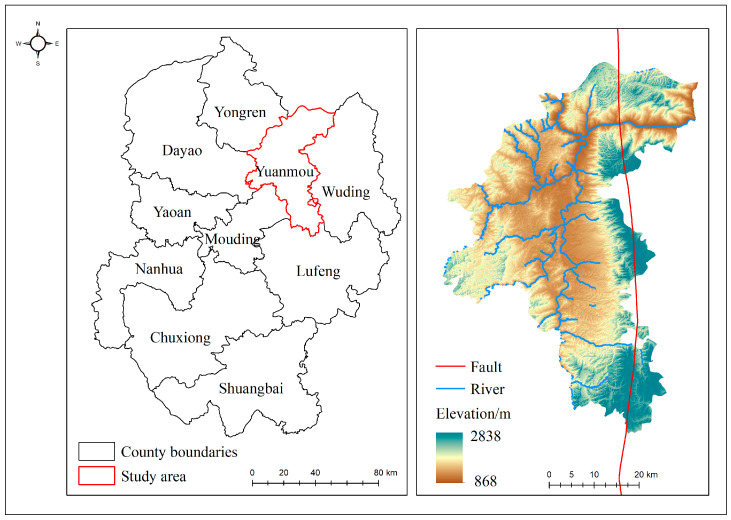
Location of the study area.

**Figure 2 sensors-24-04783-f002:**
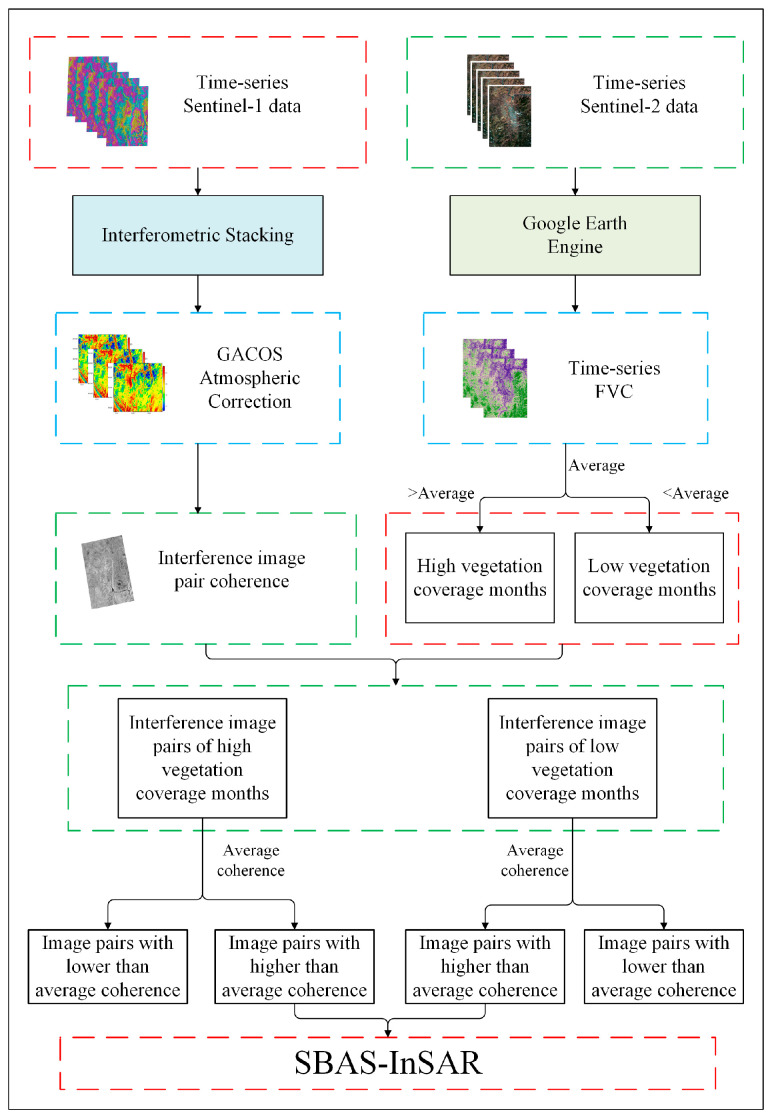
Overall technical flow chart.

**Figure 3 sensors-24-04783-f003:**
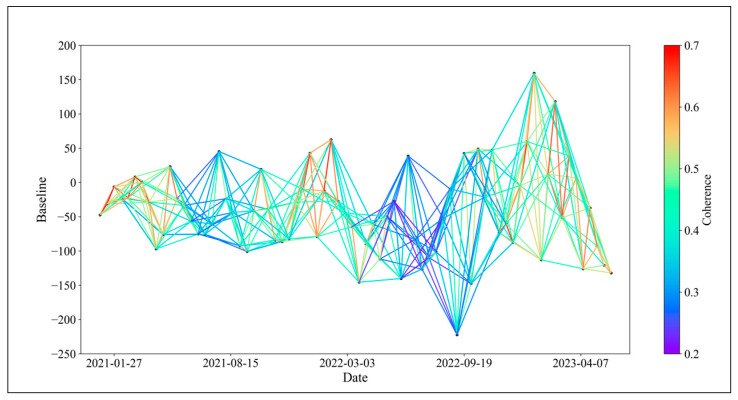
Baseline connection diagram.

**Figure 4 sensors-24-04783-f004:**
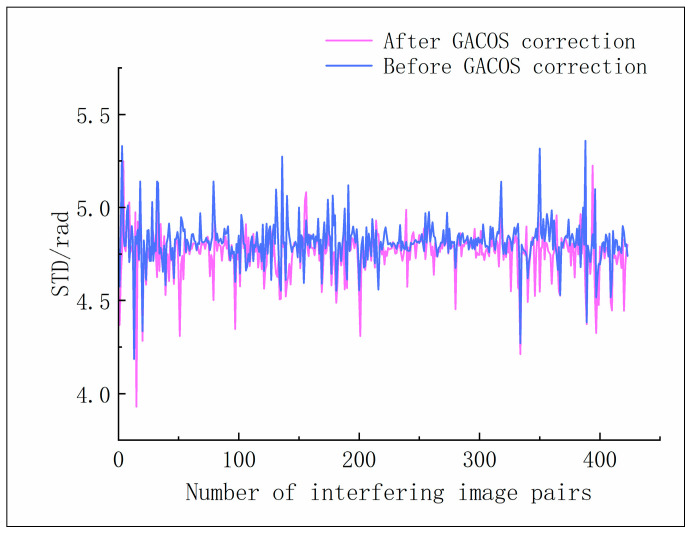
Standard deviation of the phases before and after GACOS correction.

**Figure 5 sensors-24-04783-f005:**
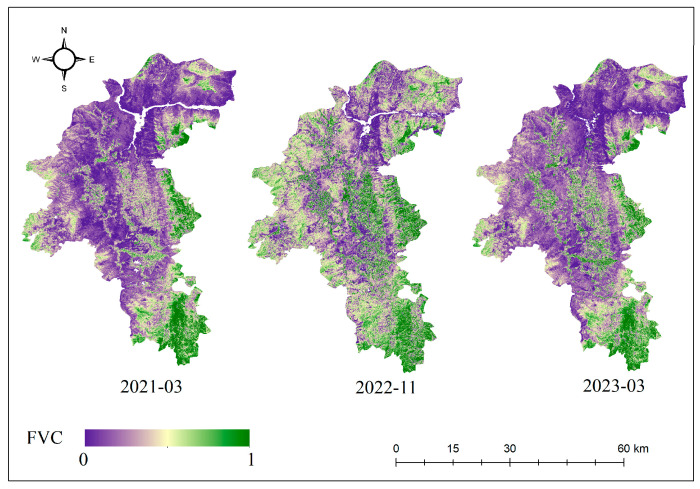
FVC in the study area.

**Figure 6 sensors-24-04783-f006:**
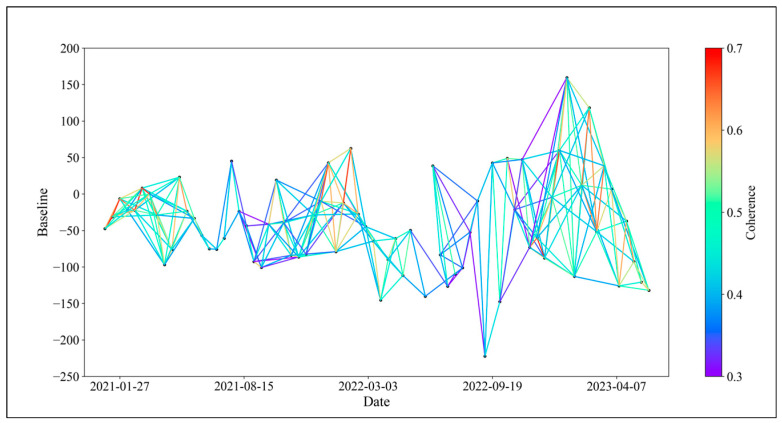
Optimized baseline connection diagram.

**Figure 7 sensors-24-04783-f007:**
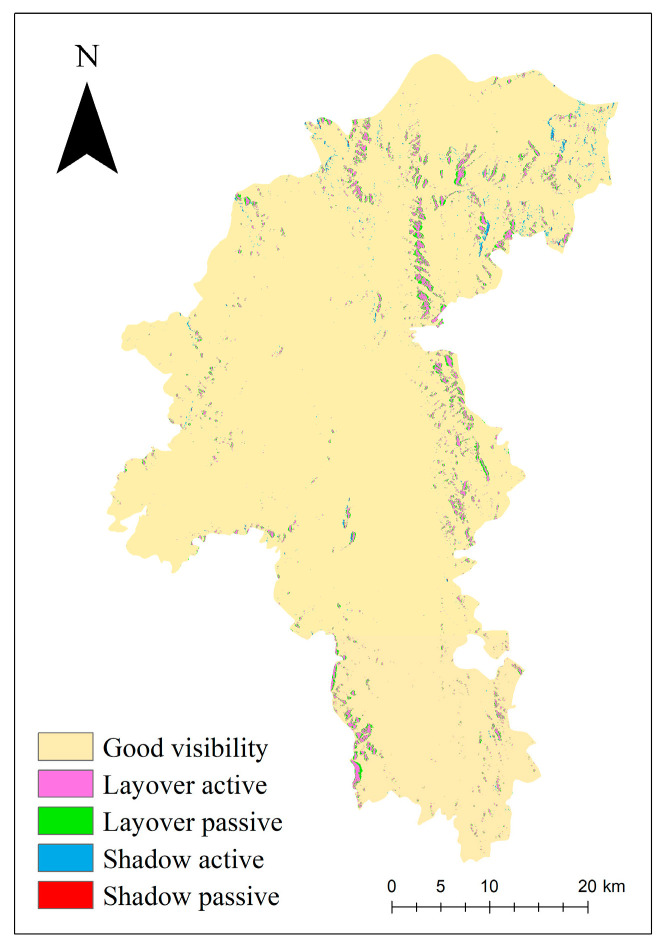
Radar visibility analysis of Yuanmou dry-hot valley.

**Figure 8 sensors-24-04783-f008:**
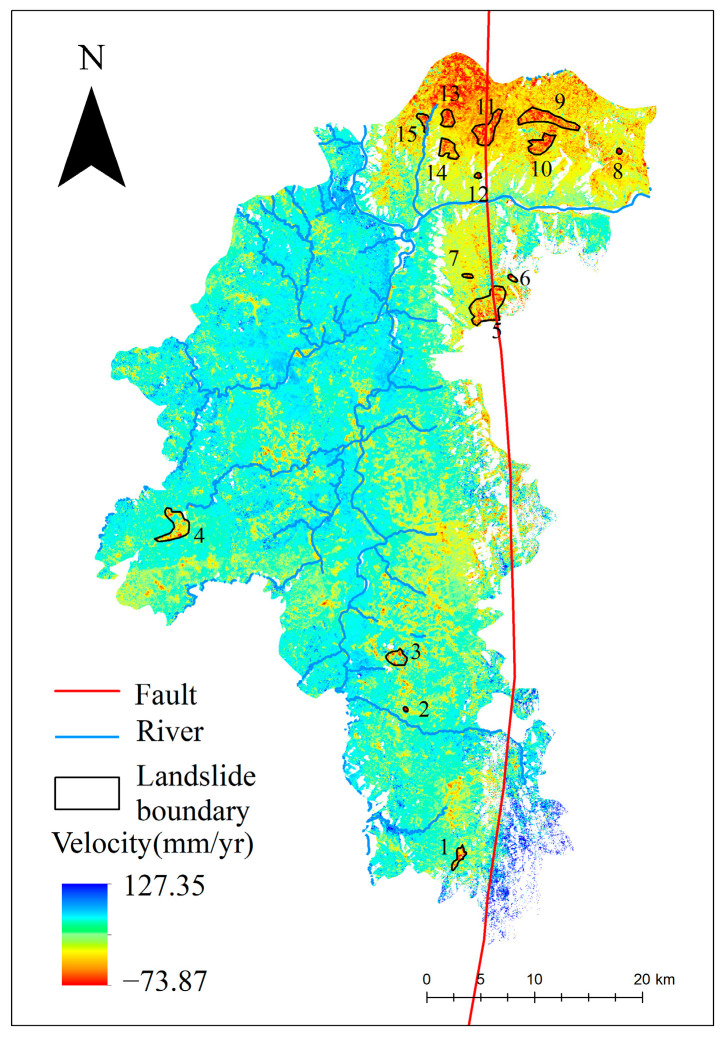
Identification results of landslide disasters in Yuanmou dry-hot valley.

**Figure 9 sensors-24-04783-f009:**
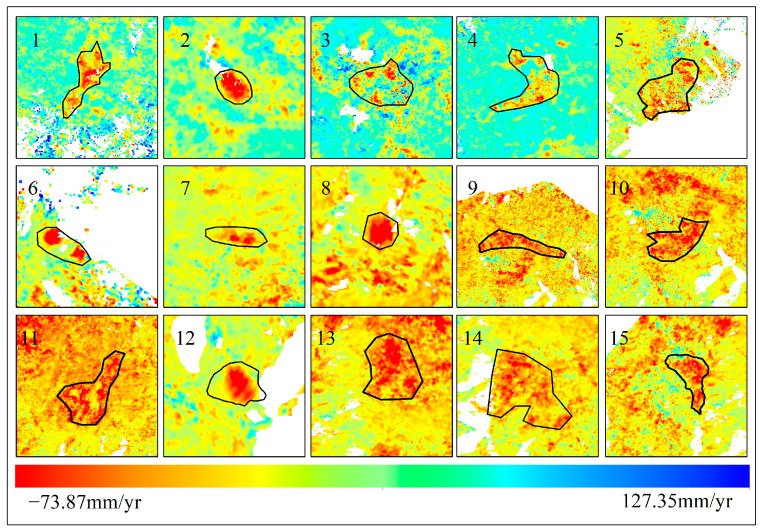
Detailed information on landslide deformation features in Yuanmou dry-hot valley.

**Figure 10 sensors-24-04783-f010:**
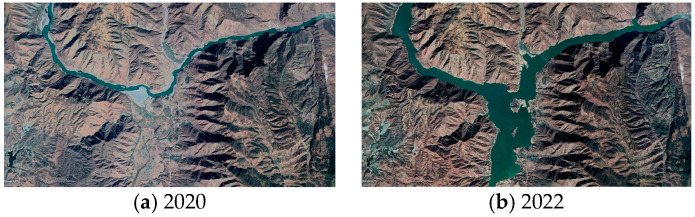
Google Earth satellite image comparison of the confluence of the Jinsha River and the Longchuan River (**a**) before the water level rises, and (**b**) after the water level rises.

**Figure 11 sensors-24-04783-f011:**
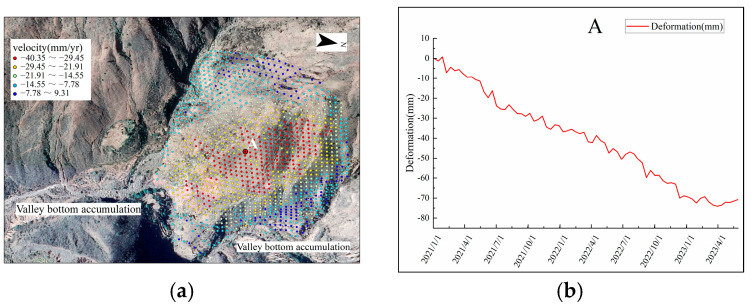
Landslide creep variant 12 (**a**) is landslide creep 12 deformation rate points (**b**) is landslide creep 12 time-series deformation curves.

**Figure 12 sensors-24-04783-f012:**
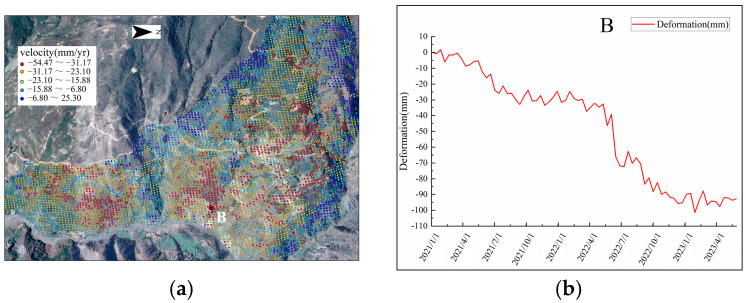
Landslide creep variant 15: (**a**) deformation rate points, and (**b**) time-series deformation curves.

**Figure 13 sensors-24-04783-f013:**
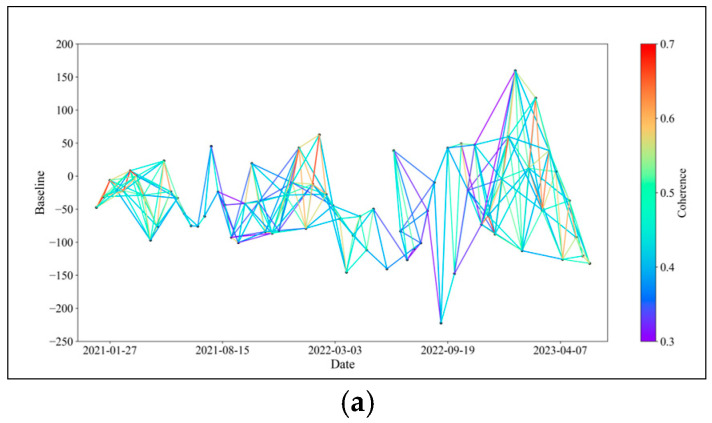
Baseline connection diagram of the following three methods: (**a**) the method of this article, (**b**) the average coherence coefficient threshold method, and (**c**) no baseline optimization.

**Figure 14 sensors-24-04783-f014:**
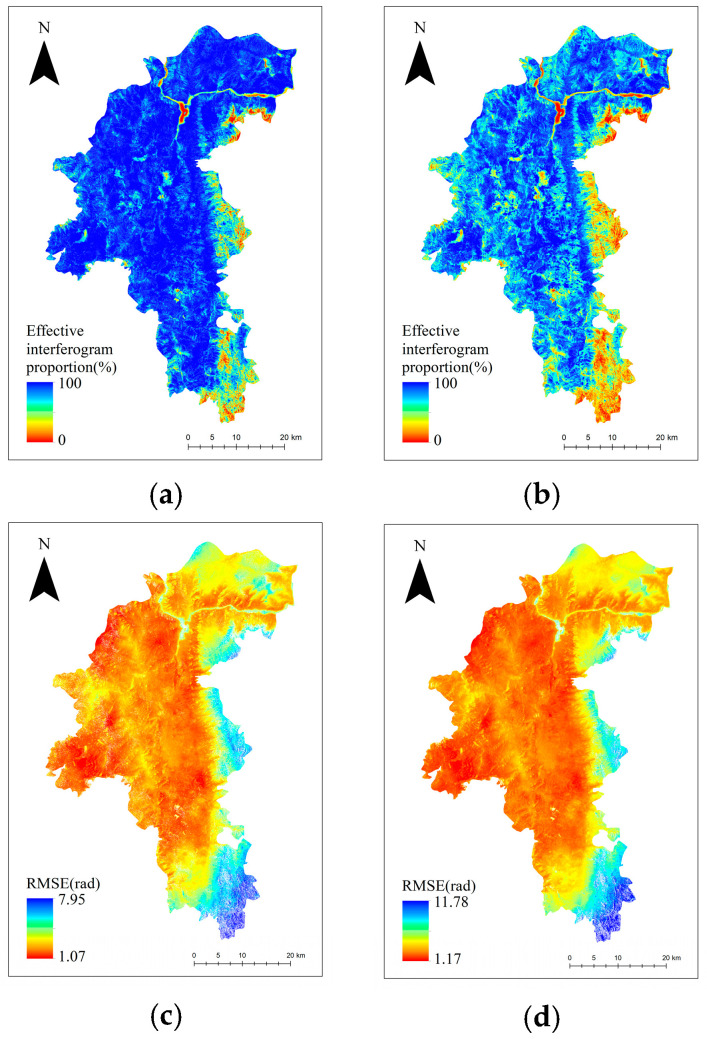
Comparison between the method in this paper and the method without baseline optimization: (**a**) the effective interferogram scale for the method in this paper; (**b**) the interferogram scale that is not optimized for baseline; (**c**) the inversion error of the method in this paper; and (**d**) is the inversion error without baseline optimization.

**Table 1 sensors-24-04783-t001:** Data description.

The Name of the Data	Phases	Resolution/m	Source
Sentinel-1A	January 2021–May 2023	5 × 20	ESA
Sentinel-2	January 2021–May 2023	10	ESA
Copernicus Sentinel POD Precision Orbit Ephemeris	January 2021–May 2023	/	ESA
DEM	January 2021–May 2023	30	JAXA
GACOS	January 2021–May 2023	90	Newcastle University
Google Satellite Imagery	January 2021–May 2023	0.2	Google Earth

**Table 2 sensors-24-04783-t002:** Effective interferogram scale and overall RMSE average.

	Effective Interferogram Scale (%)	RMSE (rad)
Method of this article	95.9	2.1
No baseline optimization method used	78.4	2.6

## Data Availability

The data used in this study include the following: ① C-band Sentinel-1A ascending orbit data from the European Space Agency (ESA) Copernicus program, consisting of 74 scenes in Interferometric Wide (IW) mode, with Single Look Complex (SLC) images in VV polarization. The spatial resolution is 5 m × 20 m, covering the time span from January 2021 to May 2023. These data were utilized for interferometric stacking to obtain surface deformation information in the Yuanmou dry-hot valley. They can be freely downloaded from https://search.asf.alaska.edu/#/. ② Optical remote sensing images from the European Space Agency (ESA) Copernicus Sentinel-2 program, with a spatial resolution of 10 m. These images were used to calculate the Fraction Vegetation Coverage (FVC) in the study area and can be freely downloaded from https://scihub.copernicus.eu. ③ Copernicus Sentinel POD Precision Orbit Ephemeris, obtained from the European Space Agency (ESA), were used to enhance satellite orbit accuracy. These data can be accessed online at https://dataspace.copernicus.eu/. ④ A 30-meter resolution Digital Elevation Model (DEM) provided by the Japan Aerospace Exploration Agency (JAXA) ALOS WORLD 3D. This DEM was used to correct terrain phase effects in Sentinel-1A data processing and can be obtained online at https://www.eorc.jaxa.jp/ALOS/en/aw3d30/data/index.htm. ⑤ Global Atmospheric Correction Online Service (GACOS) data, used to correct atmospheric delay errors in interferometric pairs generated from Sentinel-1A data stacking. These data enhance InSAR accuracy and are available for free at http://www.gacos.net/. ⑥ Google satellite imagery, utilized to assist in landslide identification based on InSAR results, can be accessed at http://www.google.cn/intl/zh-CN/earth/.

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
