# Peer review of "Study on Optimization Method for InSAR Baseline Considering Changes in Vegetation Coverage"

_sensors, 2024, doi:10.3390/s24154783_

Round 1
Reviewer 1 Report
Comments and Suggestions for Authors
The manuscript presents an optimization method for InSAR baseline, specifically addressing the impact of vegetation coverage changes on the coherence of SAR images. The method is validated in the Yuanmou dry-hot Valley, using a combination of Sentinel-1A SAR data and Sentinel-2 optical remote sensing data. The proposed OM-InSAR-BCCVC method shows significant improvements in coherence and reduction of root mean square error (RMSE) in surface deformation inversion.
1.The methodology section (3.1 to 3.3) could benefit from additional details on the specific algorithms used for coherence calculation and threshold determination. For instance, how is the average coherence coefficient specifically computed and what are the justifications for the chosen thresholds? There is a lack of a clear comparative analysis framework in section 5. While improvements are shown, the statistical significance of these improvements should be highlighted. More rigorous statistical tests or validation metrics would strengthen the claim.
2.The spatial resolution of different datasets varies, which might introduce inconsistencies in the analysis. The manuscript should address how these differences are reconciled. Figures 6 and 8, which depict baseline connection diagrams and deformation rates, respectively, lack sufficient resolution and clarity. High-resolution versions of these figures should be included.
3.The discussion in section 5 needs more depth regarding the limitations of the proposed method. For example, how does the method perform in regions with different types of vegetation or varying climatic conditions? The analysis should include more detailed error quantification and a discussion of the potential sources of error in the measurements.
4.There are several instances of awkward phrasing and minor grammatical errors that need correction for better readability. For example, in the abstract, the sentence "Significant seasonal changes in vegetation coverage in the Yuanmou dry-hot Valley lead to noticeable seasonal variations in InSAR coherence" could be rephrased for clarity.
5.To improve the literature review by incorporating recent advancements in InSAR baseline optimization techniques, including the following references can be beneficial:
An advanced scheme for range ambiguity suppression of spaceborne SAR based on blind source separation.
This paper presents a sophisticated approach to mitigating range ambiguities in spaceborne SAR data using blind source separation techniques. It is relevant to your work because range ambiguity can affect the coherence and quality of interferograms, which are critical for accurate InSAR baseline optimization.
Effective Denoising of InSAR Phase Images via Compressive Sensing
This paper discusses a method for denoising InSAR phase images using compressive sensing. The techniques for improving the signal-to-noise ratio in phase images are directly applicable to enhancing the coherence of interferometric pairs, thus benefiting InSAR baseline optimization.
Author Response
Thank you very much for your comments on our manuscript. We have responded to your questions one by one. Please see the attachment for details.
Best wishes

Reviewer 2 Report
Comments and Suggestions for Authors
Dear authors of "Study on Optimization Method for InSAR Baseline Considering Changes in Vegetation Coverage."
The study analyses the network of interferograms for vegetation cover and how atmospheric noise can be reduced for the generation of time series using the baseline and the advanced SBAS technique. In addition to being implemented, the processing of optical images for landslide identification is subsequently verified by optical remote sensing images. Sentinel 1 data are used, such as the Single Look Complex SLC product, which contains the interferometric phase, from the Copernicus mission of the European space agency. From January 2021 to May 2023, the LOS surface deformation rate in the hot dry valley in the Yuanmou study area, China. Atmospheric corrections are also carried out using GACOS, the optimization of the baseline and the masking of the geometric distortion region, which ranged from -73.87 mm/year to 127.35 mm/year. Landslides and potential landslide sites have been identified, located in area of interest. with a maximum subsidence exceeding 100 mm in two notable points. The proposed method is OM-InSAR-BCCVC, which reduces incoherent noise caused by changes in vegetation cover, optimizing the accuracy of InSAR monitoring.
This paper was reviewed chapter by chapter. It is structured in the following main sections: introduction, study area and data preparation, research method and data processing, results and analysis, discussion, and conclusion.
· The introduction addresses the main focuses and the state of the art of InSAR; it is suggested to include references on the SBAS technique, e.g.
Orellana, F., Moreno, M., & Yáñez, G. (2022). High-Resolution Deformation Monitoring from DInSAR: Implications for Geohazards and Ground Stability in the Metropolitan Area of Santiago, Chile. Remote Sensing, 14(23), 6115.
Giorgini, E., Orellana, F., Arratia, C., Tavasci, L., Montalva, G., Moreno, M., & Gandolfi, S. (2023). InSAR Monitoring Using Persistent Scatterer Interferometry (PSI) and Small Baseline Subset (SBAS) Techniques for Ground Deformation Measurement in Metropolitan Area of Concepción, Chile. Remote Sensing, 15(24).
· The Study area and data presentation section, the area overview, is well defined and includes Geographic location and topographical features, Climatic characteristics, and Vegetation features. However, I suggest dividing it into two sections such as Study area overview, and Data collection.
Line 175, In Table 1 I suggest changing the name to “data features” or “data description”
· Research Methods and Data Processing section, include: 1) interference superposition and GACOS atmospheric correction based on Sentinel-1A data, 2) pixel bipartite FVC calculation based on Sentinel-2 data, and 3) baseline optimization and SBAS-InSAR surface deformation information retrieval.
· Line 189 -193 It is clear to me that the interferogram network was optimized, but what was the criterion for selecting the interferometric phase in relation to time (days) to eliminate multitemporal incoherence? please explain.
· Line 220-225, In what way did the atmospheric correction using GACOS influence the results?
· Although the method is clear, I suggest indicating how vegetation can influence landslides, please add references in the introduction or in the methods.
· Baseline optimization and inversion of SBAS-InSAR surface deformation information, the SBAS theory has been tested in many studies, theoretically it is well defined, however I would like to know what SW you have used for SBAS InSAR, multi-temporal processing, explain...
· It would be interesting to know what were the difficulties in processing the large number of interferograms, e.g. description of the computational resources, optimization, and processing times. Please explain in the same section.
· Results and analysis, fifteen landslides and potential areas numbered from 1 to 15 are identified, with detailed information on the deformation in the study area.
· the time series that type of movement represents, is there acceleration or rather a slow movement. Please explain. Minor correction: improve the quality of the information of the time series A, underlining in figure 12 a
· The maximum displacement is considered for periods of low vegetation cover, on the contrary for periods of high cover the landslides tend to stabilize. Further prove that this is in fact true is needed.
· Line 421 - 424: How does the proposed method improve the overall coherence of the interferometric pairs and maintain the connectivity of the interferometric reference network? Please explain.
· Improve the conclusion, highlighting the scientific contribution of the method and its future developments.
Congratulations on the work, there are no major formatting errors, the figures are in order and in general of good quality, the tables are clear and easy to interpret, the paper is legible for the reader. Personally, I think it is a good contribution to Sensor journal.
Best regards.
Comments on the Quality of English LanguageThe use of English is fine. I found only minor issues worth reading carefully but no major improvements are necessary.
Author Response
Thank you very much for your comments on our manuscript. We have responded to your questions one by one. Please see the attachment for details
Best wishes
